# Effect of TiO2 Sol and PTFE Emulsion on Properties of Cu–Sn Antiwear and Friction Reduction Coatings

**Lixia Ying [1,*], Zhen Fu [1], Ke Wu [1], Chunxi Wu [1], Tengfei Zhu [1], Yue Xie [1] and Guixiang Wang [2]**

[1]  College of Mechanical and Electrical Engineering, Harbin Engineering University, Harbin 150001, China; fuzhen0826@163.com (Z.F.); kirkwuke@163.com (K.W.); wuchunxi789789@163.com (C.W.); zhutengf89115@163.com (T.Z.); xieyue@hrbeu.edu.cn (Y.X.)

[2]  College of Materials Science and Chemical Engineering, Harbin Engineering University, Harbin 150001, China; wangguixiang@hrbeu.edu.cn

[*]  Correspondence: yinglixia@hrbeu.edu.cn

**Abstract:** The aim of this paper is to obtain Cu–Sn composite coatings incorporated with PTFE and TiO2 particles, which have superior antiwear and friction reduction properties. Electrodeposition was carried out in a pyrophosphate electrolyte, and the electrochemical behavior of the plating solutions was estimated. PTFE emulsion and TiO2 sol were prepared and used, of which the average particle sizes were less than 283 and 158 nm, respectively. Then, four different types of coatings, Cu–Sn, Cu–Sn–TiO2, Cu–Sn–PTFE and Cu–Sn–PTFE–TiO2, were electroplated with a pulsed power supply. Their microstructure, composition, microhardness, corrosion resistance and tribological properties were then analyzed and compared in detail. The results show that both PTFE and TiO2 are able to improve coating structure and corrosion resistance, while they have different effects on hardness and tribological properties. However, the presence of both PTFE and TiO2 in the deposited coating leads to a lower friction coefficient of 0.1 and higher wear and corrosion resistance.

**Keywords:** electrodeposition; Cu–Sn; PTFE; TiO2 sol; tribological properties

## 1. Introduction

Copper–Tin (Cu–Sn) alloys are widely applied to various kinds of friction parts for their excellent self-lubricating properties, which can effectively reduce friction and wear under oil-free lubrication conditions [1,2]. In recent decades, electroplating Cu–Sn base coatings has attracted extensive interest. Initially, researchers focused on experimental parameters and electrolyte composition regarding enhanced properties. Various baths appeared and were used for electrodeposition of Cu–Sn alloy, such as phosphate fluoborate, boron–fluoride, pyrophosphate and cyanide based [3,4]. In addition, some new process methods were explored to enhance the self-lubricating properties of Cu–Sn composite coatings [5].

In recent years, adding self-lubricating particles to plating solutions seems to have become an effective method to further reduce the friction coefficient of coatings. It is reported that the applying of multi-walled carbon nanotubes reduces the friction coefficient and wear loss of Copper-Tin alloy to 28% and 32%, respectively [6]. That aside, as a potential lubricating material, polytetrafluoroethylene (PTFE) is usually adapted to modify compositing coatings. With the addition of PTFE to plating solutions, Balaji et al. [7] and Du et al. [8] obtained Cu–Sn–PTFE and Cu–Sn–Zn–PTFE composite coatings, respectively. Both of the coatings possessed superior self-lubricating properties. In the course of friction, the composite particles precipitated from the coating and formed a lubricating film to reduce friction and wear [9].

Evidently, the presence of PTFE soft particles lowers the friction coefficient of the Cu–Sn alloy but the shortcomings of a Cu–Sn alloy are also obvious, including softness and weak carrying capacity. More importantly, the coatings are seriously damaged under a heavy load. Therefore, it becomes more and more important to improve the hardness and wear resistance of the composite coatings and meet the requirements of a heavier load and harsh working conditions.

There are some studies with the aim of improving the composite coatings' hardness and wear resistance by co-depositing hard particles or through various electrodeposition methods [10–13]. With the addition of nano-$Al_2O_3$ to the electroless Ni–P–PTFE alloy plating solution, Xu et al. [14] developed a Ni–P–PTFE–$Al_2O_3$ composite coating with increased hardness and wear resistance. Chen et al. [15] obtained a Ni–P matrix composite coating containing nano-$Al_2O_3$ particles and PTFE particles by utilizing nano-$Al_2O_3$ and PTFE in the plating solution. The evidence shows that the incorporation of two different kinds of particles enhances the wear resistance along with reducing the friction coefficient. In fact, nano-sized particles are more prone to agglomerating in an electroplating bath. Although various surfactants were adapted in an attempt to disperse nano-particles, it was still difficult to obtain a uniformly dispersed solution, which is one of the main problems associated with fabrication of nano-composite coatings.

To overcome the non-uniformity of the dispersion particles, coating preparation was also conducted through combining a sol method with a traditional electrodeposition method by some researchers. Chen et al. added $TiO_2$ Sol into the conventional acid Ni electroplating solution to strengthen the coatings [16,17]. The research showed that the nano $TiO_2$ particles embedded in the deposited metal matrix restrain the growth of the deposited metal, leading to the formation of a more uniform and fine microstructure. Wang et al. [18,19] obtained sol-reinforced Ni–B–$TiO_2$ and Au–Ni–$TiO_2$ nano-composite coatings. Compared with Ni–B and Au–Ni coatings, respectively, their mechanical properties and wear resistance improved greatly along with the increasing nano-hardness. Based on those analyses, it can be concluded that sol is a highly dispersed system which can replace traditional powder in the plating solution to promote uniform dispersion of hard particles in the coating. Once nano-$TiO_2$ is embedded in the deposited metal layer, and restrain the growth of the deposited metal, a more uniform and fine microstructure could be achieved.

Thus, the objective of the present study is to obtain a Cu–Sn base composite coating with superior comprehensive properties. Based on this, $TiO_2$ sol and PTFE emulsion were prepared and added into the solution to codeposite with the Cu–Sn alloy. Simultaneously, the electrodepositional behavior of the bath, the particle size and dispersion state of the nanoparticles were studied. Four different kinds of Cu–Sn base coatings, Cu–Sn, Cu–Sn–PTFE, Cu–Sn–$TiO_2$ and Cu–Sn–PTFE–$TiO_2$, were obtained and their properties were evaluated and compared, especially in terms of tribological and anti-corrosion properties.

## 2. Experimental Section

### 2.1. Electroplating Processes and Methods

Stainless steel 9Cr18 (Huatai, Yangzhou, China) was chosen as a cathode substrate with dimensions of 28 mm × 25 mm × 1 mm. Prior to electroplating, the substrate was ground by using CW (CW, type of abrasive paper made in Yuli, Xianning, China) series sand paper of 1500 grade, then degreased ultrasonically in alkaline and acid solution alternately.

During electroplating, a unidirectional pulse current from Shenzhen Shicheng Electronic Technology Co., Ltd. (Shenzhen, China) was used. The main parameters involve average current ($I_a$ = 25 mA/cm$^2$), pulse frequency ($f$ = 2000 Hz) and duty cycle ($\theta$ = 60%). The corresponding current-on times ($t_{on}$), current-off times ($t_{off}$) and peak current density ($I_p$) are respectively 3 ms, 2 ms and 41.7 mA/cm$^2$. Many experiments were carried out to obtain appropriate particle concentration of PTFE emulsion and $TiO_2$ sol [20]. On the basis of previous experiments, the concentration of PTFE

and $TiO_2$ were determined at 15 and 1 g/L, respectively. The plating solution compositions and technological parameters are shown in Table 1.

**Table 1.** Plating solution compositions and technological parameters.

| Compositions/Parameters | Quantity |
|---|---|
| $K_4P_2O_7 \cdot 3H_2O$ | 260–270 g/L |
| $Cu_2P_2O_7 \cdot 4H_2O$ | 20 g/L |
| $KNaC_4H_4O_6 \cdot 4H_2O$ | 30–35 g/L |
| $Na_2SnO_3 \cdot 3H_2O$ | 40 g/L |
| $KNO_3$ | 40 g/L |
| $Na_3C_6H_5O_7 \cdot 2H_2O$ | 20 g/L |
| PTFE | 15 g/L |
| $TiO_2$ sol | 1 g/L |
| time | 1 h |
| pH | 9–10 |
| speed | 100 r/min |
| temperature | 35–40 °C |

In the plating solution, the main salt ions are $Cu^{2+}$ and $SnO_3^{2-}$. Potassium pyrophosphate ($K_4P_2O_7 \cdot 3H_2O$) is the main complexing agent, which provides $P_2O_7^{4+}$ to cause a complexation reaction with $Cu^{2+}$ and $Sn^{2+}$. This transforms the discharge ion of tin from $SnO_3^{2-}$ to $[SnP_2O_7^{4+}]^{4-}$ and promotes the co-deposition of copper and tin. Potassium sodium tartrate ($KNaC_4H_4O_6 \cdot 4H_2O$) is an auxiliary coordination agent, which can prevent the precipitation of copper hydroxide and the hydrolysis of stannate. Simultaneously, with relatively positive discharge potential, potassium nitrate ($KNO_3$) as a depolarizing agent can effectively reduce the polarization of the cathode and significantly promote tin deposition. Similarly, sodium citrate ($Na_3C_6H_5O_7 \cdot 2H_2O$) is an additive, which can also reduce the cathode polarization and indent the deposition potential of $Cu^{2+}$ and $Sn^{4+}$.

The concentrations of PTFE and $TiO_2$ during deposition in each plating solution are shown in Table 2. In the bath solution, the $Cu^{2+}$, $Sn^{4+}$, PTFE and $TiO_2$ particles all have a positive charge. Under the influence of an electric field force, the particles move towards the cathode phase and are adsorbed on the cathode material before, finally, the $Cu$–$Sn$–PTFE–$TiO_2$ coating forms. The formation process is shown in Figure 1.

**Table 2.** The type and quantity of composite particles in each plating solution.

| Content | Cu–Sn | Cu–Sn–PTFE | Cu–Sn–TiO$_2$ | Cu–Sn–PTFE–TiO$_2$ |
|---|---|---|---|---|
| PTFE (g/L) | 0 | 15 | 0 | 15 |
| TiO$_2$ (g/L) | 0 | 0 | 1 | 1 |

**Figure 1.** The schematic diagram of the Cu–Sn–PTFE–TiO$_2$ coating formation process (note that the relative sizes of ions/atoms/particles are not to scale).

*2.2. Characterization of Plating Solution and Cu–Sn–PTFE–TiO$_2$ Composite Coatings*

A 380ZLS model laser particle size/potential meter (Particle Sizing Systems, Port Richey, FL, USA) was used to measure the size of the TiO$_2$ sol and PTFE particles. In addition, the plating solutions of Cu–Sn–PTFE, Cu–Sn–TiO$_2$ and Cu–Sn–PTFE–TiO$_2$ were observed by transmission electron microscopy (TEM, FEI, Hillsboro, OR, USA). The samples for the TEM observation were made in the following way: take a drop of the solution using a plastic dropper and spread it on a micro-grid of carbon film on a copper mesh, then heat it to bake off the alcohol and water using an electric baking lamp. The surface morphologies and compositions of coatings were investigated with scanning electron microscopy (SEM, FEI) and X-ray Fluorescence (XRF, PANalytical, Almelo, the Netherlands). Hardness of the coatings, an average of five indentations, was measured by Vickers Hardness Tester HVS-1000 (Jujing, Shanghai, China) with a load of 2 N for 15 s on the surface.

Tribological properties of coatings were characterized by a friction–abrasion testing machine in room temperature. The friction counterpart was a GCr15 steel ball of 5 mm in diameter. A load of 100 g and a stage rotated speed of 200 r/min were used with a wear time of 10 min for each sample. Finally, the corrosion resistance of coatings was analyzed by Tafel polarization tests, which were conducted in a three-electrode system. The counter electrode was a platinum slice, and the reference electrode was a standard calomel electrode (SCE). The tests were performed in 3.5% NaCl electrolyte with an electrochemical workstation (model CHI660B, Chenhua, Shanghai, China) at a scan rate of 2 mV/s. The electrochemical behavior of the plating solution was also tested with this workstation. The volt–ampere characteristic curves were obtained by linear sweep voltammetry (LVS).

## 3. Results and Discussion

*3.1. Analysis of Plating Solution*

### 3.1.1. Electrodepositional Behavior of Basic Solution

Figure 2 shows the volt–ampere characteristic curves (LSV) of the solutions, including K$_4$P$_2$O$_7$. The LSV of the solution with just K$_4$P$_2$O$_7$·3H$_2$O (260 g/L) is shown in Figure 2a, which shows the current rising gradually with the potential negative shift. Although hydrogen evolution starts to occur when the potential goes negative to −1.6 V, there is no other reduction peak during the scanning range (−0.2∼−1.6 V) and the solution exhibits good stability. The LSV of the solutions with Cu$_2$P$_2$O$_7$·4H$_2$O (20 g/L) are shown in Figure 2b. It can be seen that the cathode current starts to rise at −0.4 V due to the reduction of Cu$^{2+}$ ions to Cu metal (curve a in Figure 2b). With the addition of K$_4$P$_2$O$_7$·3H$_2$O (260 g/L), cathode potential of Cu shifts to approximately −1.0 (curve b in Figure 2b).

The LSV of Sn deposition is shown in Figure 2c. With Na$_2$SnO$_3$·3H$_2$O (40 g/L) and K$_4$P$_2$O$_7$·3H$_2$O (260 g/L) in the solution, the deposition of Sn$^{2+}$ ions to Sn metal starts to occur at −1.25 V. With the addition of K$_4$P$_2$O$_7$·3H$_2$O (260 g/L), two changes occur: a more positive potential −1.0 V and an increased peak. So, with K$_4$P$_2$O$_7$·3H$_2$O in the solution, the deposition of Sn seems to become easy. Reduction peak is observed at −0.9 V from the LSV of the solution containing K$_4$P$_2$O$_7$·3H$_2$O (260 g/L), Cu$_2$P$_2$O$_7$·4H$_2$O (20 g/L) and Na$_2$SnO$_3$·3H$_2$O (40 g/L), shown in Figure 2d. In contrast with Figure 2b,c, the gap between the reduction potentials of Cu and Sn is found to be diminished with the presence of K$_4$P$_2$O$_7$·3H$_2$O. Therefore, the inclusion of K$_4$P$_2$O$_7$·3H$_2$O facilitates the co-deposition by lowering the difference in reduction potentials of the two individual metals (see Figure 2d).

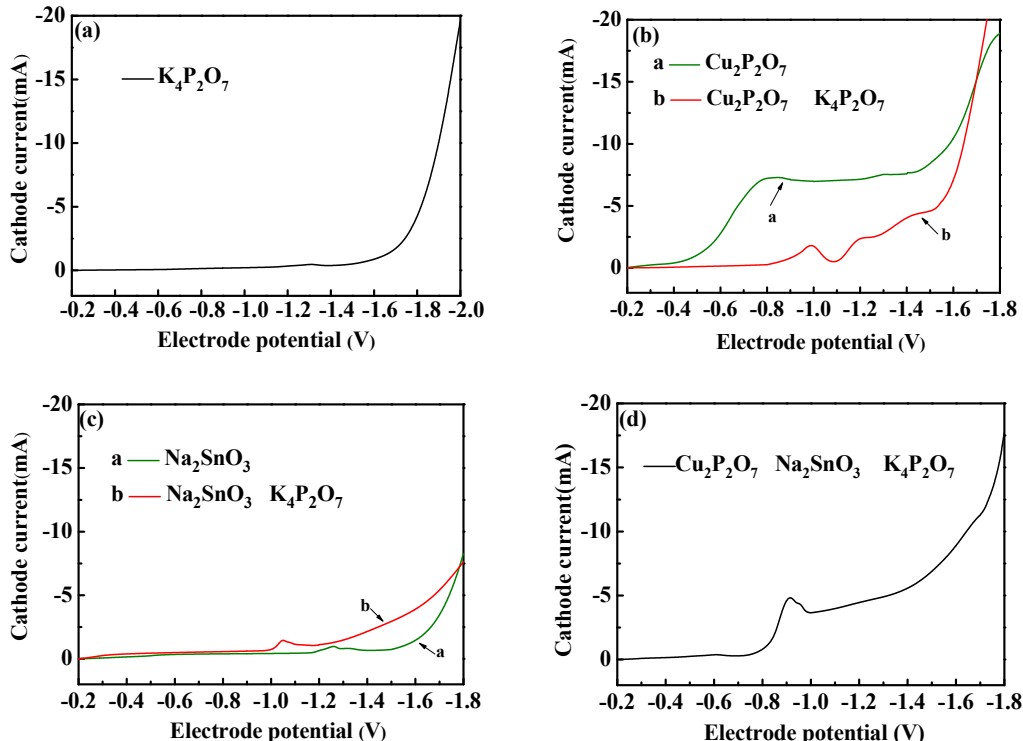

**Figure 2.** The volt–ampere characteristic curves of solutions: (**a**) $K_4P_2O_7$ solution; (**b**) electrolytic deposition of $Cu^{2+}$; (**c**) electrolytic deposition of $Sn^{4+}$; (**d**) electrolytic deposition of $Cu^{2+}$ and $Sn^{4+}$.

### 3.1.2. Analysis of $TiO_2$ and PTFE particles

Figure 3 shows the distribution ranges of particle sizes and average diameters of the PTFE and $TiO_2$. While both particle size distributions show log-normal distributions (as expected), the left cut-off in Figure 3a (PTFE particles) is caused by the detection limit of the particle analyzer. In other words, particles < 25 mm could not be detected. Thus, it can be concluded that the average particle diameters of PTFE are less than 283 nm. Figure 3b shows that the average particle diameters of $TiO_2$ is 158 nm.

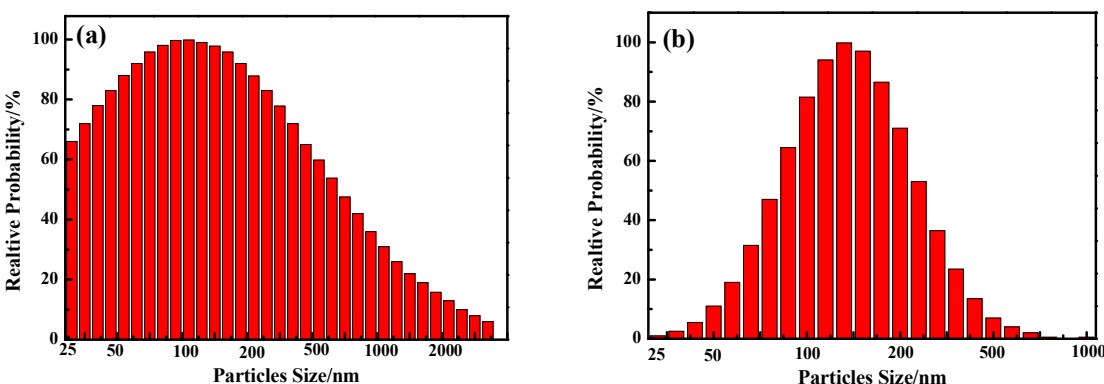

**Figure 3.** The size distribution of PTFE emulsion and $TiO_2$ sol: (**a**) PTFE emulsiom; (**b**) $TiO_2$ sol (note that the detection limit of the particle analyzer is 25 mm).

The obtained PTFE emulsion and $TiO_2$ sol particles were added into the base plating solution for further testing. With the addition of 15 g/L PTFE and 1 g/L $TiO_2$, respectively, the particles were uniformly dispersed in the plating solution and no agglomeration occurred, as observed in Figure 4a,b. Additionally, the particle size was not significantly changed (compared with Figure 3). When PTFE

and TiO$_2$ were added to the solution together, the solution remained steady and the dispersity of the nanoparticles was still well (Figure 4c).

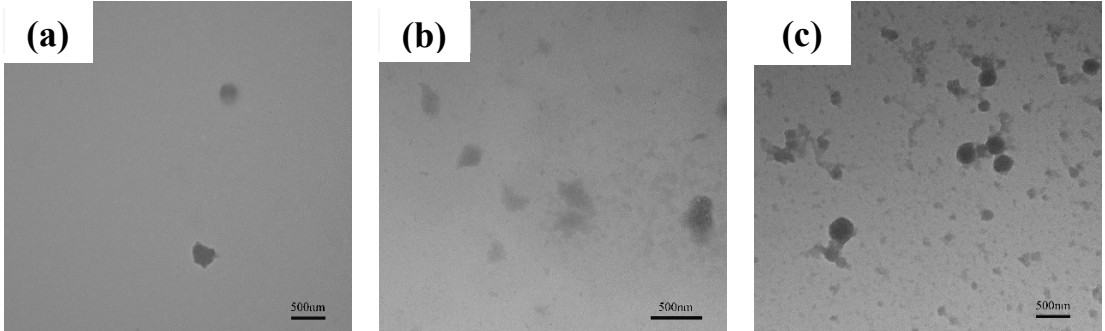

**Figure 4.** The dispersion state morphologies of PTFE emulsion and TiO$_2$ sol in electroplating solution: (**a**) PTFE emulsion; (**b**) TiO$_2$ sol; (**c**) PTFE–TiO$_2$.

*3.2. Microstructure and Composition of the Coatings*

The SEM micrographs of coatings, including Cu–Sn, Cu–Sn–PTFE, Cu–Sn–TiO$_2$ and Cu–Sn–PTFE-TiO$_2$ composite coatings are shown in Figure 5. According to SEM results, the surface of each coating is uniform and there are no such defects as pinhole, cracks, drain plating, etc. With the addition of TiO$_2$, the coatings of Cu–Sn–TiO$_2$ and Cu–Sn–PTFE–TiO$_2$ are smoother than Cu–Sn and Cu–Sn–PTFE. Indeed, the results also show TiO$_2$ sol has dramatically effect on improving the coating structure than that of PTFE emulsion. However, with PTFE and TiO$_2$ coexisting in the coating, Cu–Sn–PTFE–TiO$_2$ composite coating possess the smoothest surface and finest structure.

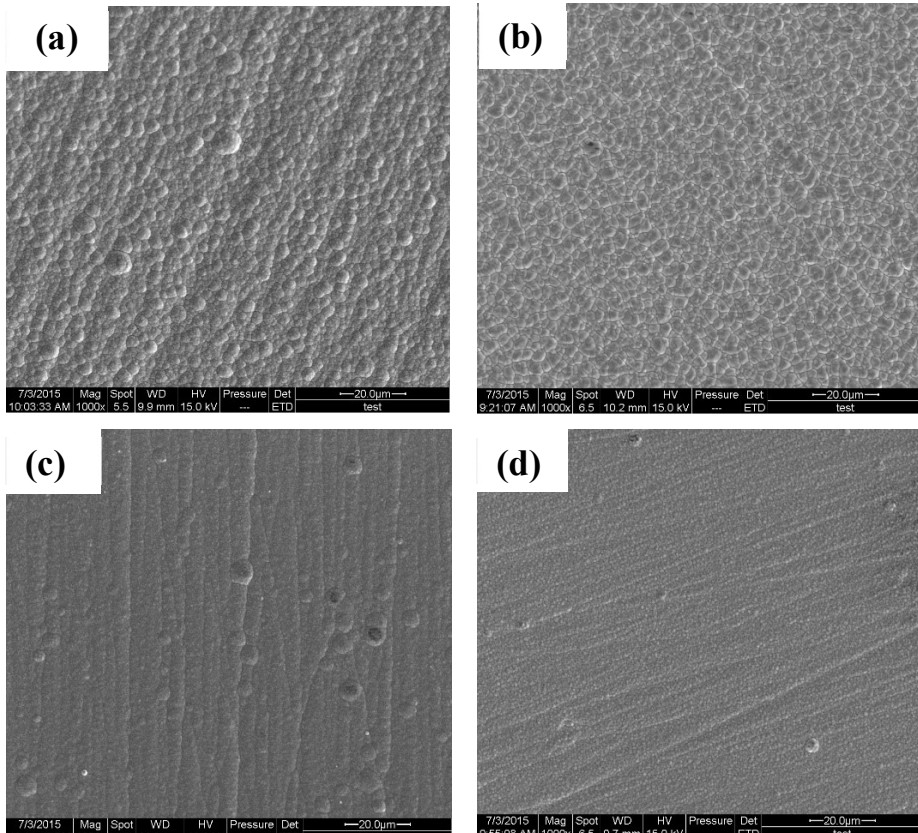

**Figure 5.** The SEM morphologies of Cu–Sn base coatings: (**a**) Cu–Sn; (**b**) Cu–Sn–PTFE; (**c**) Cu–Sn–TiO$_2$; (**d**) Cu–Sn–PTFE–TiO$_2$.

Table 3 shows the composition of the deposits produced with different particles in the bath. According to the XRF analysis, the alloy composition is dependent on the composition of the bath. Although both of the contents of Cu and Sn decrease with the addition of PTFE and TiO$_2$, the proportions of Cu/Sn in the coatings stay basically unchanged. Composition analysis also confirms that PTFE and TiO$_2$ coexist in the coating of Cu–Sn–PTFE–TiO$_2$.

**Table 3.** The compositions of the coatings.

| Conversion Coatings | Cu (wt %) | Sn (wt %) | PTFF (wt %) | TiO$_2$ (wt %) |
|---|---|---|---|---|
| Cu–Sn | 92.42 | 7.58 | — | — |
| Cu–Sn–PTFE | 87.21 | 6.68 | 6.11 | — |
| Cu–Sn–TiO$_2$ | 90.71 | 6.76 | — | 1.54 |
| Cu–Sn–PTFE–TiO$_2$ | 86.84 | 6.46 | 5.50 | 1.20 |

### 3.3. Corrosion Properties

The potentiodynamic polarization curves of different kinds of Cu–Sn base composite coatings are shown in Figure 6. Table 4 presents the corresponding corrosion current density and corrosion potential. The corrosion resistances of Cu–Sn–PTFE, Cu–Sn–TiO$_2$ and Cu–Sn–PTFE–TiO$_2$ coatings are much better than that of the Cu–Sn alloy coating in 3.5% NaCl solution. Cu–Sn–PTFE–TiO$_2$ composite coating also has the highest corrosion potential and minimum corrosion current density, which are $-0.256$ V and 1.443 A/cm$^2$, respectively.

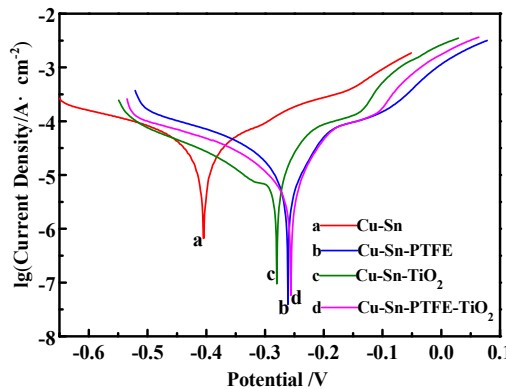

**Figure 6.** Potentiodynamic polarization curves of different coatings.

**Table 4.** Potentiodynamic polarization parameters of coatings.

| Coatings | Corrosion Potential/V | Corrosion Current Density/A·cm$^{-2}$ |
|---|---|---|
| Cu–Sn | $-0.405$ | 4.067 |
| Cu–Sn–PTFE | $-0.261$ | 1.856 |
| Cu–Sn–TiO$_2$ | $-0.280$ | 2.434 |
| Cu–Sn–PTFE–TiO$_2$ | $-0.256$ | 1.443 |

Figure 7 presents the corroded surface topographies of four kinds of coatings. The main corrosion form of all of them was point corrosion in 3.5% NaCl electrolyte. However, the depth of the corrosion pits was significantly different in those coatings. It is obvious that Cu–Sn–PTFE–TiO$_2$ has the shallowest corrosion pits and the smoothest surface.

In any case, with the same corrosion medium and corrosion time, the anti-corrosion performances of Cu–Sn–PTFE, Cu–Sn–TiO$_2$, Cu–Sn–PTFE–TiO$_2$ coatings were distinctly better than that of Cu–Sn alloy coatings. Presumably the main reason is that the pulses co-deposited of PTFE and TiO$_2$ nanoparticles produce fine grains, which strengthen matrix structure, compact the microstructure and improve the corrosion resistance. In addition, PTFE and TiO$_2$ nanoparticles can fill in the intergranular

pores of matrix grain, and the reduction of the pore size on the material surface can prevent the corrosion ions getting through micropores of the composite material, so the anti-corrosion performance is effectively improved (Table 4).

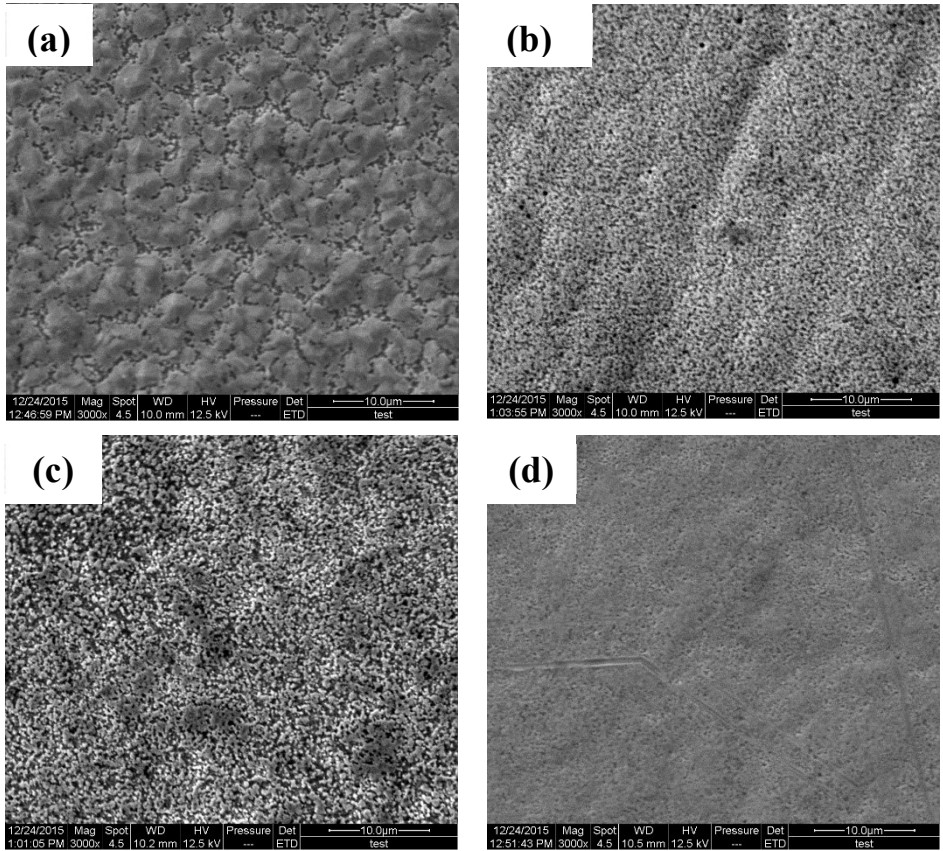

**Figure 7.** The SEM morphologies of corroded coatings: (**a**) Cu–Sn; (**b**) Cu–Sn–PTFE; (**c**) Cu–Sn–TiO$_2$; (**d**) Cu–Sn–PTFE–TiO$_2$.

*3.4. Mechanical Properties*

3.4.1. Hardness of Coatings

The hardness values of different kinds of Cu–Sn base composite coatings are listed in Table 5. The results suggest that the hardness of Cu–Sn–TiO$_2$, Cu–Sn–PTFE–TiO$_2$, Cu–Sn and Cu–Sn–PTFE coatings rank from high to low, indicating that TiO$_2$ nanoparticles are able to increase the hardness of the coating, while PTFE particles decrease. In the electrodeposition process, the hard TiO$_2$ nanoparticles were embedded in the Cu–Sn alloy matrix, which lead to higher hardness than Cu–Sn coating. That could presumably be interpreted as fine-grain strengthening and dispersion strengthening effect. However, due to the softness of PTFE particles, hardness of the Cu–Sn–PTFE coating is lower than that of Cu–Sn coating. With the synergistic effect of TiO$_2$ and PTFE, the Cu–Sn–PTFE–TiO$_2$ composite coating can achieve higher hardness than Cu–Sn coating.

**Table 5.** Hardness of the coatings.

| Coating | Hardness/HV |
| --- | --- |
| Cu–Sn | $413 \pm 4$ |
| Cu–Sn–PTFE | $375 \pm 6$ |
| Cu–Sn–TiO$_2$ | $485 \pm 6$ |
| Cu–Sn–PTFE–TiO$_2$ | $465 \pm 5$ |

### 3.4.2. Tribological Properties

The friction coefficient curves of different kinds of those coatings are presented in Figure 8. It indicates that under dry friction conditions, the Cu–Sn alloy coating exhibits the maximum friction coefficient along with poor wear resistance. According to Figure 8, PTFE decreases the friction coefficient to 0.08, but cannot improve the wear life. On the other hand, with the addition of nano-TiO$_2$, wear-resisting time of the Cu–Sn–TiO$_2$ composite coating increases, while the friction coefficient is about 0.14 between those of Cu–Sn and Cu–Sn–PTFE. However, the Cu–Sn–PTFE–TiO$_2$ composite coating has much longer wear-resisting time, as same as Cu–Sn–TiO$_2$. In addition, its friction coefficient of 0.1 is the lowest among the four kinds of coatings. Therefore, the Cu–Sn–PTFE–TiO$_2$ coating achieves the best wear-resisting and anti-friction performance with PTFE and TiO$_2$ coexisting.

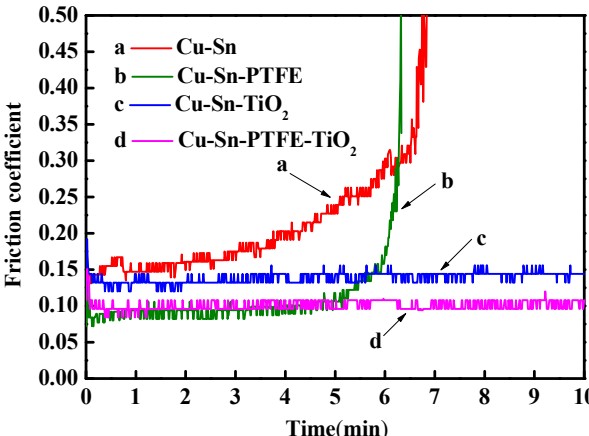

**Figure 8.** The friction coefficient curve of each Cu–Sn base plating coating.

The SEM images of the worn surfaces of those composite coatings are shown in Figure 9. Severe plastic deformation happens on the worn surface of the Cu–Sn alloy and Cu–Sn–PTFE coating (Figure 9a,b), which is usually related to adhesive wear mechanism, while narrow scratches implying abrasive wear mechanism are observed in the worn surface of the Cu–Sn–TiO$_2$ and Cu–Sn–PTFE–TiO$_2$ coatings (Figure 9c,d). Simultaneously, with the addition of TiO$_2$ (Figure 9c,d), wear tracks become smoother and their widths also become narrower than those of coatings without TiO$_2$ (Figure 9a,b). However, just some slight furrowing or scratches appear on the surface of the Cu–Sn–PTFE–TiO$_2$ composite coating, which indicate the best wear resistance (Figure 9d).

The wear mechanisms of electroplated coatings against steel can be explained in Figure 10. Steel balls with a higher hardness of HV800 were used as counterparts. When a Cu–Sn deposit with lower hardness was chosen as the tested sample, shown in Figure 10a, the small steel ball would be easily embedded in the coating matrix in the process of relative sliding, which made the contact area and friction force increase and result in serious wear damage. Once the tested sample was changed to a Cu–Sn–PTFE composite coating, as shown in Figure 10b, the friction shear force was significantly reduced with the formation of PTFE self-lubricating film, however, wear resistance of the coating was not improved due to the low hardness of the coating.

In contrast, a relatively high hardness brought by the reinforcement of TiO$_2$ nanoparticles increases the resistance of plastic distortion in the process of relative motion, plus no self-lubricating film on the surface, which leads to a higher friction coefficient (Figure 10c). However, with PTFE and TiO$_2$ coexisting in the Cu–Sn–PTFE–TiO$_2$ composite coating, as illustrated in Figure 10d, lower shear force and harder matrix correspondingly lead to a lower coefficient and higher wear resistance. That is to say, the synergistic effect of PTFE and TiO$_2$ makes the Cu–Sn–PTF–TiO$_2$ composite coating present a good wear-resisting and anti-friction performance.

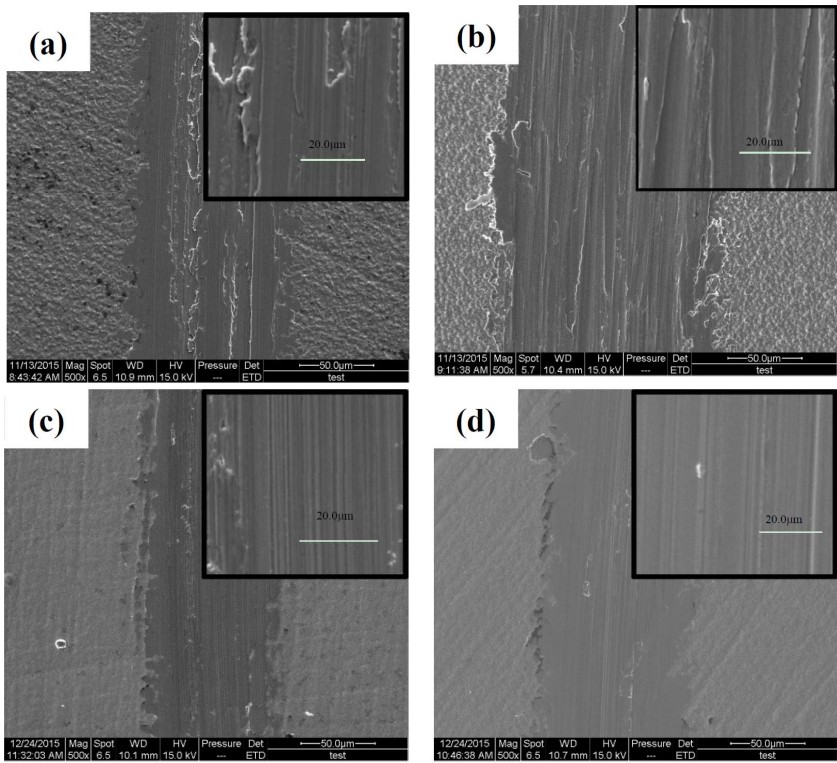

**Figure 9.** The SEM morphologies of wear traces on the surface of coatings: (**a**) Cu–Sn; (**b**) Cu–Sn–PTFE; (**c**) Cu–Sn–TiO$_2$; (**d**) Cu–Sn–PTFE–TiO$_2$.

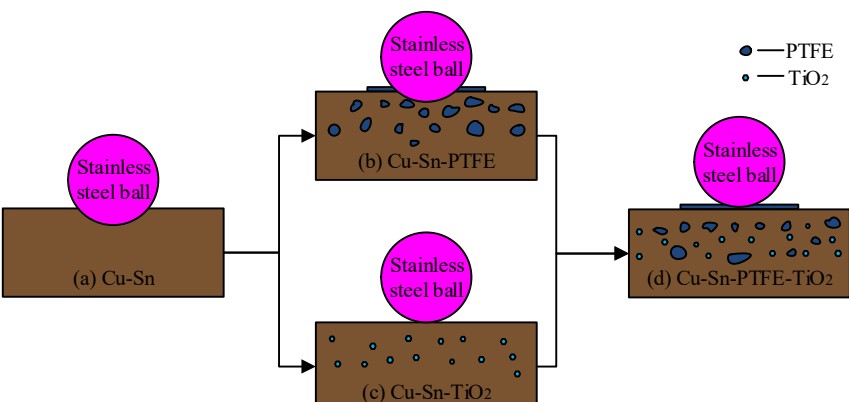

**Figure 10.** The schematic diagram of the friction mechanism of each Cu–Sn base coatings: (**a**) Cu–Sn; (**b**) Cu–Sn–PTFE; (**c**) Cu–Sn–TiO$_2$; (**d**) Cu–Sn–PTFE–TiO$_2$.

## 4. Conclusions

In the paper, PTFE emulsion and TiO$_2$ sol with average particle diameters <283 nm and ~158 nm, respectively, were successfully prepared and dispersed. TEM analysis indicates that no agglomeration occurs in the plating solution with appropriate particle concentrations. The analysis of the electrochemical behavior suggests that Cu and Sn can be co-deposited relatively easily with the addition of K$_4$P$_2$O$_7$. Hence, Cu–Sn composite coatings containing PTFE and TiO$_2$ are obtained from a pyrophosphate plating solution, which has superior antiwear and friction reduction properties.

The investigations also indicate that PTFE and TiO$_2$ nanoparticles are able to promote the coating structure and smoothness. Simultaneously, with the addition of nanoparticles, the corrosion resistance of Cu–Sn–PTFE, Cu–Sn–TiO$_2$ and Cu–Sn–PTFE–TiO$_2$ composite coatings is better than that of Cu–Sn



alloy coating. Cu–Sn–PTFE–TiO$_2$ composite coating also has the highest corrosion potential and minimum corrosion current density, which are −0.256 V and 1.443 A/cm$^2$, respectively.

In addition, the hardness of the Cu–Sn–TiO$_2$ coating is 485 HV, while that of the Cu–Sn–PTFE–TiO$_2$ coating is 465 HV due to the softness of PTFE. However, with the synergistic effect of PTFE and TiO$_2$, a Cu–Sn–PTFE–TiO$_2$ composite coating exhibits superior wear-resisting and anti-friction performance with the friction coefficient of 0.10.

**Author Contributions:** Conceptualization, L.Y.; Methodology, L.Y. and K.W.; Validation, K.W., Z.F. and C.W.; Formal Analysis, Z.F., K.W.; Investigation, Z.F., K.W.; Data Curation, L.Y., Z.F. and K.W.; Writing–Original Draft Preparation, Z.F. and K.W.; Writing–Review & Editing, L.Y., C.W., T.Z., Y.X. and G.W.

**Funding:** This research was funded by the National Nature Science Foundation of China (No.51305090) and the Fundamental Research Funds for the Central Universities (HEUCFP201814).

**Conflicts of Interest:** The authors declare no conflict of interest. The funders had no role in the design of the study; in the collection, analyses, or interpretation of data; in the writing of the manuscript, and in the decision to publish the results.

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
