# Peer review of "Effect of TiO2 Sol and PTFE Emulsion on Properties of Cu–Sn Antiwear and Friction Reduction Coatings"

_coatings, doi:10.3390/coatings9010059_

Round 1
Reviewer 1 Report
This paper presents interesting results on the synergistic effects of adding PTFE and TiO2 particles to Cu-Sn electrodeposits for wear and antifriction applications. While the results are very good and worth publishing in Coatings, the paper in its present form is not acceptable and would require major revisions.
1) The biggest problem with the paper is that critical sections are vey difficult to follow and comprehend because the English (style, grammar, choice of words) is very poor.
2) All acronyms and abbreviations must be explained in full when they are first used in the text. (e.g. PTFE, MSA, CW series, LSV, etc.).
3) More detail must be provided in the Experimental section.
a) It should be specifically stated that pulse plating was used. Current-on and current-off times should be listed together with peak current density.
b) While the Cu and Sn salts are the sources for Cu and Sn ions, the role of the rest of the bath ingredients is not clear. This should be explained. Why 31.6 g/L and not 32 g/L?
c) PTFE was added as 15 g/L of PTFE powder. However, TiO2 was added with 50 ml/L. However, it is not known how much TiO2 (in grams) is in 1 ml. Therefore the TiO2 concentration should also be given in g/L.
d) The schematic diagram in Figure 1 needs to be modified to reflect the actual sizes of particles, ions in the solution and atoms in the deposit. What are the little orange dots in the deposit?
e) It is stated that the plating solutions were observed by TEM. What was in the TEM? The actual solution in a wet sample holder? Or the dried particles. This needs to be clarified.
4) In the Results and Discussion section the following points need clarification.
a) For the particle size distribution in Figure 3a the left tail is cut-off at 25 nm. Why is that?
b) The text describing the SEM micrographs shown in Figure 5 mentions “grain size” of the various deposits. The grain sizes should be given.
c) The compositions of the deposits are given as wt% of F or Ti. This is rather meaningless. The numbers should be converted to give the PTFE and TiO2 wt% in the deposits. Also, it is not clear what the Cu and Sn concentrations were in the different deposits.
d) The hardness values for the different deposits (Table 5) should be rounded off and the standard deviations should be added.
Author Response
Dear Reviewer:
Thanks for your comments. We have revised the manuscript based on your comments.
This paper presents interesting results on the synergistic effects of adding PTFE and TiO2 particles to Cu-Sn electrodeposits for wear and antifriction applications. While the results are very good and worth publishing in Coatings, the paper in its present form is not acceptable and would require major revisions.
1) The biggest problem with the paper is that critical sections are very difficult to follow and comprehend because the English (style, grammar, choice of words) is very poor.
Response: Indeed, there are many problems about the language in the original manuscript. Thank you very much. The English, including the style, grammar, choice of words, have been revised in the entire manuscript, especially in the red paragraphs. Because a lot of revisions have been made, it is difficult to point out one by one.
2) All acronyms and abbreviations must be explained in full when they are first used in the text. (e.g. PTFE, MSA, CW series, LSV, etc.).
Response: All acronyms and abbreviations have been explained in full when they are first used in the text. e.g. polytetrafluoroethylene (PTFE) in line 41. CW (Model of Abrasive paper made in China) in line 84. linear sweep voltammetry (LVS) in line131. MSA is ethane sulfonic acid, It has been deleted in the revision, considering its’ unimportantance for the manuscript.
3) More detail must be provided in the Experimental section.
a) It should be specifically stated that pulse plating was used. Current-on and current-off times should be listed together with peak current density.
Response: The power supply is from Shenchen Shicheng Electronic Technology co., LTD. (GKPT-FB4). Unidirectional pulse current was used in this paper. The average current (Ia=25 mA cm-2), pulse frequency (f=2000 HZ) and duty cycle(θ=60%) are given. According to the following formula, ton and toff time can be calculated.
Ia= f= θ=
The calculation results are as follows: ton=3 ms, toff=2 ms,Ip=41.7 mA /cm-2
This description has been added in line87-90 (highlighted in yellow).
b) While the Cu and Sn salts the sources for Cu and Sn ions, the role of the rest of the bath ingredients is not clear. This should be explained. Why 31.6 g/L and not 32 g/L?
Response: In the plating solution, the main salt ions are Cu2+ and . Potassium pyrophosphate(K4P2O7·3H2O) as the main complexant agent, provides to cause complexation reaction with Cu2+ and Sn2+, transforms the discharge ion of tin from to [ ]4-, which is conducive to the co-deposition of copper and tin. Potassium sodium tartrate (KNaC4H4O6·4H2O) is an auxiliary coordination agent, which can prevent the precipitation of copper hydroxide and the hydrolysis of stannate. On the other hand, with relatively positive discharge potential, potassium nitrate (KNO3) as depolarizing agent can effectively reduce the polarization of the cathode and significantly promote tin deposition. Similarly, citric acid (Na3C6H5O7·2H2O) is an additive, which can also reduce the cathode polarization and indent the deposition potential of Cu2+ and Sn4+.
This description has been added in line95-104 (highlighted in yellow).
About the concentration of KNaC4H4O6·4H2O, it should be given in a range indeed. 31.6g/L is approximately the middle value. When we edit the original manuscript, the specific value used in the experiment was given. Considering scientificity, the Table.1 has been checked and complemented. (highlighted in yellow)
c) PTFE was added as 15 g/L of PTFE powder. However, TiO2 was added with 50 ml/L. However, it is not known how much TiO2 (in grams) is in 1 ml. Therefore the TiO2 concentration should also be given in g/L.
Response: TiO2 is added to the plating solution in the form of TiO2 sol. ml/L is actually the unit of TiO2 sol, and no conversion was done in the original manuscript.
According to the preparation method of TiO2 sol, conversion was carried out in the revised draft. The concentration of TiO2 in the electroplating solution is about 1g/L through calculation. 1g/L has replaced the 40ml/L in the revision (highlighted in yellow.)
d) The schematic diagram in Figure 1 needs to be modified to reflect the actual sizes of particles, ions in the solution and atoms in the deposit. What are the little orange dots in the deposit?
Response: The little orange dots represent the carbon atoms. In the original manuscript, in order to vividly display the structure of the matrix, the little orange dots were pointed. However, because the matrix structure is complex and contains multiple elements, it cannot be completely drawn. Therefore, the little orange dots have been deleted from the revision. (See Fig.1)
In the schematic diagram, the true proportion of each atom should be shown. But the diameters of the titanium dioxide and PTFE molecules are about a thousand times larger than that of iron, copper, tin and other atoms. So, in the figure, titanium dioxide and PTFE atoms only can be indicated as larger than other atoms, and cannot to be drawn specifically.
e) It is stated that the plating solutions were observed by TEM. What was in the TEM? The actual solution in a wet sample holder? Or the dried particles. This needs to be clarified.
Response: The samples for the TEM observation were made in the following way: take a drop of the solution using a plastic dropper and spread it on a micro-grid of carbon film on a copper mesh, then heated it to bake off the alcohol and water in the solution using an electric baking lamp. This description has been added in line116-1118 (highlighted in yellow).
4) In the Results and Discussion section the following points need clarification.
a) For the particle size distribution in Figure 3a the left tail is cut-off at 25 nm. Why is that?
Response: The size of the sol particles was measured by a 380ZLS model laser particle size/potential meter. The left side of the picture shows that particles less than 25nm are almost undetectable (see the original figure below). In order to easy to understand, we add the left tail in the revision. In the revised draft, we slightly modified Figure 3a. (shown in the revision)
b) The text describing the SEM micrographs shown in Figure 5 mentions “grain size” of the various deposits. The grain sizes should be given.
Response: Figure 5 shows the original morphology of the coating. The grain size was estimated from the roughness and feature of the surface, and no further measure was conducted on the grain size. We have modified the expression in the original manuscript. We will carry out further measure and more-refined analyses in the next study, including the grain size. This modification has been added in line180-186 (highlighted in yellow).
c) The compositions of the deposits are given as wt% of F or Ti. This is rather meaningless. The numbers should be converted to give the PTFE and TiO2 wt% in the deposits. Also, it is not clear what the Cu and Sn concentrations were in the different deposits.
Response: The F and Ti compositions indeed need to convert. Some coverting have been done, and the concentrations PTFE and TiO2 are obtained. And the Cu and Sn concentrations are also given in the revision. (seeTable3, highlighted in yellow)
The description/discussion has been added to the revised manuscript (see lines184-188, highlighted in yellow).
d) The hardness values for the different deposits (Table 5) should be rounded off and the standard deviations should be added.
Response: Hardness values do need to be rounded off. The corrections have been made and the standard deviations have been added (See Table 5.) The related section has also been revised (highlighted in yellow).
We have also made editorial effort to improve the language of the manuscript. We would like to thank you again for reviewing the manuscript. Hope that the revised version is satisfactory.

Reviewer 2 Report
Please add citation in line 37.
Please check for spelling and grammar mistakes, expressions, and present/past of verbs throughout the manuscript.
Line 64, please explain "nano effect" in detail.
Line 97, please specify "how much"
Line 121 and below, EWD measurements with Kelvin probe. How do you get absolute values? What kind of reference surface is used or what kind of referencing?
Figure 7, why did you choose a column representation of EWD data for different films? The four data can be just given as values in the text. Differences are not so large, so you have basically a value of 4.96 eV of the bar alloy and a change of 60 mV for all studied coatings. The differences between the coatings are not statistically significant.
In summary, I appreciate reading your manuscript which represents a well-performed experimental study, including a series of microscopic data as well as a set of macroscopic material parameters, combined with well-illustrated interpretations.
Author Response
Dear Reviewer:
Thanks for your comments. We have revised the manuscript.
-Please add citation in line 37.
Response: The citations have been added (see the Line37 and the references, highlighted in yellow).
-Please check for spelling and grammar mistakes, expressions, and present/past of verbs throughout the manuscript.
Response: The English, including spelling and grammar mistakes, expressions, and present/past of verb have been revised in the entire manuscript, especially in the red paragraphs. Because a lot of revisions have been made, it is difficult to point out one by one.
-Line 64, please explain "nano effect" in detail.
Response: The "nano effect" mentioned in the manuscript refers that TiO2 particles have small particle diameter, small size and high surface energy. Expect them embeded into the deposited metal crystal, inhibit the growth of the deposited metal, produce a more uniform and fine microstructure, and improve the hardness and wear resistance of the coating.
-Line 97, please specify "how much"
Response: In the original manuscript, TiO2 was added into the plating solution in the form of TiO2 sol. ml/L is actually the unit of TiO2 sol, and no conversion was done in the original manuscript.
Conversion is carried out in the revised draft. According to the preparation method of TiO2 sol, the concentration of TiO2 is about 1g/L when the sol in the electroplating solution are 40ml/L. (see in Table2, highlighted in yellow)
-Line 121 and below, EWD measurements with Kelvin probe. How do you get absolute values? What kind of reference surface is used or what kind of referencing?
Response: Sorry. The unit of the EWF is wrong in the original manuscript. It should be ‘’eV“,not “V”. The reference surface is gold. A gold tip of 1 mm in diameter was used to scan the surface.
-The oscillation frequency of the SKP tip was 173 Hz. Figure 7, why did you choose a column representation of EWD data for different films? The four data can be just given as values in the text. Differences are not so large, so you have basically a value of 4.96 eV of the bar alloy and a change of 60 mV for all studied coatings. The differences between the coatings are not statistically significant.
Response: We agree with your comment. The column representation of EWD data is not suitable. And the differences between the coatings are not statistically significant. At the same time, we also think that the work function measurements do not provide any additional information compared to the polarization curves. Instead they lead to confusion. So, we decide to remove the part of EWF. (see the revision)
We have also made editorial effort to improve the language of the manuscript. We would like to thank you again for reviewing the manuscript. Hope that the revised version is satisfactory.

Round 2
Reviewer 1 Report
The manuscript has been improved significantly in terms of technical content. Most of the issues raised by this reviewer in the first review have been addressed. However, there are still a few technical problems that need attention.
1) Figure 1 has been improved. However, the size issue of atoms/ions/particles remained. Perhaps a sentence could be added to the Figure caption: “Note that the relative sizes of ions/atoms/particles are not the scale.”
2) Table 1 shows that sodium citrate was added to the solution. However, the text (line 102/103) says citric acid. This should be changed.
3) The problem with Figure 3 is still the same. While both particle size distributions show log-normal distributions (as expected), the left cut-off in Figure 3a (PTFE particles) has not been explained. Is this because the particle size analyzer is not reliable at particles<25mm? An explanation must be given. Perhaps a sentence: “Note that the detection limit of the particle analyzer is 25mm” in the figure caption would do the trick.
However, if this is the case, giving the average particle size as 2.82.8nm (line 158) is meaningless. The same is true for the conclusions (line 279) and abstract (line 15). I suggest the average particle sizes should be listed as ~158nm and<280nm, respectively.
Author Response
Dear Reviewer:
Thanks for your comments. We have revised the manuscript.
The manuscript has been improved significantly in terms of technical content. Most of the issues raised by this reviewer in the first review have been addressed. However, there are still a few technical problems that need attention.
1) Figure 1 has been improved. However, the size issue of atoms/ions/particles remained. Perhaps a sentence could be added to the Figure caption: “Note that the relative sizes of ions/atoms/particles are not the scale.”
Response: Thanks for your suggestion. A sentence has been added to the Figure caption: “Note that the relative sizes of ions/atoms/particles are not the scale.” (see line99-100, highlighted in yellow).
2) Table 1 shows that sodium citrate was added to the solution. However, the text (line 102/103) says citric acid. This should be changed.
Response: Citric acid has been instead by sodium citrate in the text (line 91, highlighted in yellow).
3) The problem with Figure 3 is still the same. While both particle size distributions show log-normal distributions (as expected), the left cut-off in Figure 3a (PTFE particles) has not been explained. Is this because the particle size analyzer is not reliable at particles<25mm? An explanation must be given. Perhaps a sentence: “Note that the detection limit of the particle analyzer is 25mm” in the figure caption would do the trick.
However, if this is the case, giving the average particle size as 282.8nm (line 158) is meaningless. The same is true for the conclusions (line 279) and abstract (line 15). I suggest the average particle sizes should be listed as ~158nm and<283nm, respectively.
Response: Yes. Indeed, the detection limit of the particle analyzer is 25mm.
The explanation has been added in line144-146.
A sentence has been added to the Figure caption: “Note that the detection limit of the particle analyzer is 25mm”. (see line152, highlighted in yellow)
The average particle sizes of TiO2 and PTFE have been listed as ~158nm and<283nm, respectively. Anywhere they appear in the manuscript, we have revised. (see line14, 146, 258, highlighted in yellow)
Another main concern remains. While the English has been improved, the paper would benefit enormously from yet another round of editing for English. Perhaps an independent editing service would help.
Response: The languages about the introduction and analysis have been reorganized. The English, including style, grammar, choice of words, have been revised in the entire manuscript, especially in the red paragraphs.
Revisions have been made very much, so that it is difficult to point out one by one.
We have also made editorial effort to improve the language of the manuscript. We would like to thank you again for reviewing the manuscript. Hope that the revised version is satisfactory.

Reviewer 2 Report
The paper is acceptable.
Round 3
Reviewer 1 Report
The paper is now acceptable for publication in Coatings.